# Exponential Topology-enabled Scalable Communication in Multi-agent Reinforcement Learning

**Xinran Li**[1,2]  **Xiaolu Wang**[3]*  **Chenjia Bai**[2]  **Jun Zhang**[1]
[1]The Hong Kong University of Science and Technology
[2]Institute of Artificial Intelligence (TeleAI), China Telecom
[3]Software Engineering Institute, East China Normal University
`xinran.li@connect.ust.hk, xiaoluwang@sei.ecnu.edu.cn`
`baicj@chinatelecom.cn, eejzhang@ust.hk`

## Abstract

In cooperative multi-agent reinforcement learning (MARL), well-designed communication protocols can effectively facilitate consensus among agents, thereby enhancing task performance. Moreover, in large-scale multi-agent systems commonly found in real-world applications, effective communication plays an even more critical role due to the escalated challenge of partial observability compared to smaller-scale setups. In this work, we endeavor to develop a scalable communication protocol for MARL. Unlike previous methods that focus on selecting optimal pairwise communication links—a task that becomes increasingly complex as the number of agents grows—we adopt a global perspective on communication topology design. Specifically, we propose utilizing the exponential topology to enable rapid information dissemination among agents by leveraging its small-diameter and small-size properties. This approach leads to a scalable communication protocol, named ExpoComm. To fully unlock the potential of exponential graphs as communication topologies, we employ memory-based message processors and auxiliary tasks to ground messages, ensuring that they reflect global information and benefit decision-making. Extensive experiments on large-scale cooperative benchmarks, including MAgent and Infrastructure Management Planning, demonstrate the superior performance and robust zero-shot transferability of ExpoComm compared to existing communication strategies. The code is publicly available at `https://github.com/LXXXXR/ExpoComm`.

## 1 Introduction

Cooperative multi-agent reinforcement learning (MARL) has recently emerged as a promising approach for complex decision-making tasks across diverse real-world applications, such as resource allocation (Ying & Dayong, 2005), package delivery (Seuken & Zilberstein, 2007), autonomous driving (Zhou et al., 2021), robot control (Swamy et al., 2020), and infrastructure management planning (Leroy et al., 2024). Under the widely adopted centralized training and decentralized execution (CTDE) paradigm (Kraemer & Banerjee, 2016; Lyu et al., 2021), algorithms like MADDPG (Lowe et al., 2017), COMA (Foerster et al., 2018), MATD3 (Ackermann et al., 2019), QMIX (Rashid et al., 2020), and MAPPO (Yu et al., 2022) have achieved notable success.

To enhance agent collaboration in partially observable scenarios, communication mechanisms have been incorporated into multi-agent systems (MASs) to assist in decentralized decision-making (Sukhbaatar et al., 2016). Enabling information exchange during execution helps MARL algorithms to address non-stationarity and partial observability prevalent in these environments. Building upon this foundation, researchers have devoted efforts to designing effective communication protocols, focusing on three core considerations: 1) *whom* the agents should communicate with (Ding et al., 2020; Hu et al., 2024); 2) *when* communication should occur (Hu et al., 2021; Kim et al., 2019); and 3) *how* the agents should design and utilize the communication messages

---

*Corresponding author.

effectively (Das et al., 2019; Guan et al., 2022). By leveraging tools such as attention and graph neural networks (GNNs), learnable and adaptive communication mechanisms have significantly advanced MARL performance.

Despite considerable success, most existing communication strategies are designed for small-scale MASs (Lowe et al., 2017; Samvelyan et al., 2019; Peng et al., 2021) and may struggle as systems scale to dozens or even hundreds of agents, which are ubiquitous in real-world applications (Cui et al., 2022; Schmidt et al., 2022; Yang et al., 2023; Ma et al., 2024). In these *many-agent* systems, existing methods that learn pairwise connectivity among agents falter for two reasons: First, these methods often require agents to receive messages only from "useful" peers. However, identifying these peers becomes increasingly challenging as the number of agents grows, potentially compromising the effectiveness of communication protocols (Guan et al., 2022). Second, the overhead of these methods scales poorly. Specifically, training memory consumption quickly becomes prohibitively large, as shown in our empirical evaluation, and the communication overhead during execution scales quadratically with the number of agents, which is infeasible for many-agent systems.

This motivates a fundamental rethinking of scalable MARL communication: Can we adopt a global perspective and design an overall topology that propagates information among all agents effectively and at low cost, rather than relying on finding task-specific pairwise connectivity? In this vein, we propose an exponential topology-enabled communication protocol, termed *ExpoComm*, as a scalable solution for MARL communication. Unlike previous works that seek to identify useful communication links at each timestep, ExpoComm draws inspiration from graph theory and leverages the small-diameter property of exponential topologies to ensure effective communication by facilitating message flow across all agents within a limited number of timesteps. The inherent sparsity (small size) of exponential topologies allows ExpoComm's communication cost to scale (near-)linearly with the number of agents. Moreover, to fully leverage the small-size and small-diameter properties of exponential graphs for efficient information dissemination, we employ memory-based blocks for message processing and auxiliary tasks to ground messages, ensuring that they effectively reflect global information. Extensive experiments across twelve scenarios on large-scale benchmarks, including MAgent (Zheng et al., 2018) and Infrastructure Management Planning (IMP) (Leroy et al., 2024), demonstrate the superior performance of ExpoComm compared to baseline algorithms when handling large numbers of agents up to a hundred. Additionally, owing to its global perspective without pairwise reliance, ExpoComm exhibits remarkable zero-shot transferability to larger numbers of agents during test time.

## 2 RELATED WORK

**Communication in MASs** Communication among agents in MARL was first introduced by Sukhbaatar et al. (2016); Foerster et al. (2016) and has since become an active research area due to its potential to enhance cooperation and improve task performance. The flexibility of communication protocols makes finding effective solutions for the MARL paradigm challenging (Zhu et al., 2024). To address this difficulty, many studies have focused on optimizing communication components, such as message generators, message aggregators, and connectivity among agents, through end-to-end training (Peng et al., 2017). From the sender side, ToM2C (Wang et al., 2022) and MAIC (Yuan et al., 2022) enhance message generation through teammate modeling, while CACL (Lo et al., 2024) uses contrastive learning techniques to learn communication encoding in a decentralized training paradigm. From the receiver side, TarMAC (Das et al., 2019), G2ANet (Liu et al., 2020), and MASIA (Guan et al., 2022) improve message aggregation using attention-based strategies.

Recently, researchers have addressed challenges posed by real-world communication systems. Notably, NDQ (Wang et al., 2020b) and TMC (Zhang et al., 2020) reduce communication costs by crafting succinct messages, while ATOC (Jiang & Lu, 2018), IC3 (Singh et al., 2019), I2C (Ding et al., 2020), and CommFormer (Hu et al., 2024) manage overhead by pruning unnecessary communication links. Additionally, Freed et al. (2020) propose a stochastic encoding/decoding scheme to handle noisy channels, and DACOM (Yuan et al., 2023) introduces delay-aware communication to account for the high latency of wireless channels.

Despite these advancements, scalability in communication mechanisms has been largely overlooked, often due to the quadratically increasing communication cost associated with fully-connected graphs as the number of agents grows. Although few works explicitly address the scalability issue, efforts to

design communication topologies among agents offer potential solutions. These can be categorized into fully-connected, rule-based, and learned topologies. Early works (Sukhbaatar et al., 2016; Foerster et al., 2016; Peng et al., 2017) typically adopt fully-connected topologies to demonstrate communication benefits, but at the cost of high bandwidth requirements. Later on, to reduce the overall communication overhead, Jiang et al. (2020) and Weil et al. (2024) restrict communication to nearby neighbors based on distance, while NeurComm (Chu et al., 2020) limits communication to neighboring agents in networked MASs. In spite of achieving significant performance gains, their further applicability may be limited since they require extra information beyond local observation to determine the communication topology. In contrast, learned topology methods assume no such requirements and offer high flexibility. In particular, ATOC (Jiang & Lu, 2018), IC3 (Singh et al., 2019), I2C (Ding et al., 2020) locally deploy gates for agents to decide if they should engage in communication. However, these methods may result in uncontrollable overall communication costs due to individual control schemes. Alternatively, MAGIC (Niu et al., 2021) utilizes graph attention mechanisms to learn the communication topology, while CommFormer (Hu et al., 2024) extends the idea and enables control over the overall communication sparsity. Although effective in small-scale MASs, peer-wise connectivity becomes increasingly difficult to learn in large-scale MASs, and high sparsity may impair performance, as discussed by Hu et al. (2024).

Our proposed ExpoComm, which incorporates rule-based topologies for rapid information dissemination among all agents, complements existing efforts in MAS communication by explicitly addressing scalability challenges.

**Exponential Graphs** Exponential graphs are a class of graph topologies that exhibit strong scalability properties with respect to the number of nodes. They have been primarily used in distributed learning to periodically synchronize model updates across devices. Assran et al. (2019) investigate exponential graphs with gossip algorithms and achieve high consensus rates for decentralized learning. Follow-up works (Wang et al., 2020a; Ying et al., 2021; Kong et al., 2021; Yuan et al., 2021) build upon this topology, optimizing model weight update algorithms and providing empirical evidence and theoretical guarantees for the effectiveness of exponential graphs. Beyond distributed learning, exponential graphs also have applications in chip design (Wang et al., 2015; 2016). Overall, exponential graphs demonstrate efficient information dissemination across many nodes, making them a promising candidate topology for achieving scalable communication in MARL.

## 3 Scalable Communication with Exponential Graph in MARL

In this section, we propose ExpoComm, which leverages exponential graphs as communication topologies among agents in MARL to enable scalable communication. We structure the following subsections to address three key questions: 1) Why and how should exponential graphs be adapted for agent communication? 2) How can the corresponding neural network architecture be designed to effectively utilize the messages transmitted through these topologies? 3) How can messages propagated among agents be grounded to ensure their usefulness?

In Section 3.1, we outline the requirements for scalable communication: effective information dissemination among agents and low communication overhead. We translate these requirements into the challenge of identifying topologies with small diameters and sizes, key properties of exponential topologies. In Section 3.2, we discuss how memory-based message processors can enable meaningful message encoding, leveraging the small-diameter property over multiple timesteps within exponential topologies. In Section 3.3, we adopt a global perspective to ground messages using a global state reconstruction auxiliary task and contrastive learning, as ExpoComm aims to facilitate message flow across the entire graph rather than focusing on local features.

### 3.1 Exponential Graph as the Communication Topology

#### 3.1.1 Problem Setting

In this work, we consider a fully cooperative partially observable multi-agent task, which can be modeled as a decentralized partially observable Markov decision process (Dec-POMDP) (Oliehoek & Amato, 2016). The Dec-POMDP is defined by a tuple $\mathcal{M} = \langle \mathcal{S}, A, P, R, \Omega, O, N, \gamma \rangle$ with $N$ being the number of agents and $\gamma \in (0, 1]$ being the discount factor. At each timestep $t$, with the

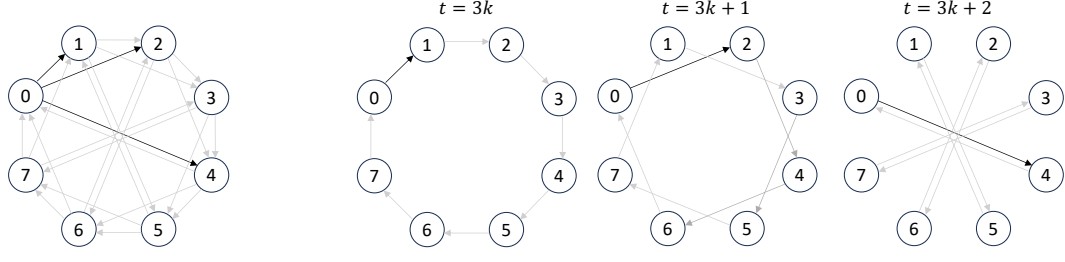

(a) Static exponential graph.

(b) One-peer exponential graph. $k$ is an integer.

Figure 1: Illustration of exponential graphs with $N = 8$.

global observation $s^t \in \mathcal{S}$, agent $i$ receives a local observation $o_i^t \in \Omega$ and then communicates with other agents. Upon receiving the messages from other agents, agent $i$ then selects an action $a_i^t \in A$ based on its local policy $\pi_i$. These individual actions collectively form a joint action $\boldsymbol{a}^t \in A^N$, leading to a transition to the next global observation $s^{t+1} \sim P(s^{t+1} | s^t, \boldsymbol{a}^t)$ and inducing a global reward $r^t = R(s^t, \boldsymbol{a}^t)$. The team objective is to learn the policies that maximize the expected discounted cumulative return $G_t = \sum_t \gamma^t r^t$.

### 3.1.2 COMMUNICATION TOPOLOGIES

To design an effective and scalable communication protocol in many-agent systems, it is essential to determine whom to communicate with, i.e., to construct the communication topology so that communication is both beneficial for decision-making and cost-effective. While previous work (Hu et al., 2024) assumes a static communication topology, we adopt a more flexible, time-varying directed graph $\mathcal{G}^t = \langle \mathcal{V}, \mathcal{E}^t \rangle$, where node $v_i \in \mathcal{V}$ denotes agent $i$ and edge $e_{i \to j}^t \in \mathcal{E}^t$ indicates a communication link from agent $i$ to agent $j$ at timestep $t$.

From a graph perspective, we consider the following desiderata for the communication topology:

- **Small graph diameter for fast information dissemination**: Formally defined as diameter$(\mathcal{G}^t) = \max_{v_i, v_j \in \mathcal{V}} d(v_i, v_j)$ with $d(v_i, v_j)$ representing the shortest path distance from $v_i$ to $v_j$, the graph diameter indicates how quickly messages travel through the graph. Since communication aids multi-agent decision-making by providing the locally observant agents with global information and alleviating the non-stationarity, a graph with a small diameter can expedite message exchange and is therefore desirable.

- **Small size for low communication overhead**: Formally defined as $|\mathcal{E}^t|$, the size of a graph denotes the total number of edges, corresponding to the number of communication links in an MAS. We assume that any message transmission incurs the same overhead, therefore the total overhead scales with the number of links. Given the high hardware requirement for communication modules and the potential delays induced by densely connected communication topologies, we prefer graphs with a small size in many-agent settings.

### 3.1.3 EXPONENTIAL GRAPHS

Based on the desiderata above for the communication topologies, we draw inspiration from graph literature and choose exponential graphs (Assran et al., 2019; Ying et al., 2021) as a promising candidate for communication topology in many-agent systems. Below, we introduce two variants of exponential graphs and demonstrate their small-diameter and small-size properties through an illustrative example.

**Static Exponential Graph** Assuming a randomly sequential ordering of agents $0, 1, \dots, N-1$ and the corresponding adjacency matrix $E \in \{0, 1\}^{N \times N}$, in the static exponential graph, each agent communicates with peers that are $2^0, 2^1, \dots, 2^{\lfloor \log_2(N-1) \rfloor}$ hops away, which is illustrated by Figure 1a. Formally, we have

$$E_{ij}^{t(\text{stat})} = \begin{cases} 1 & \text{if } \log_2\left((j-i) \bmod N\right) \text{ is an integer or } i = j \\ 0 & \text{otherwise} \end{cases}. \tag{1}$$

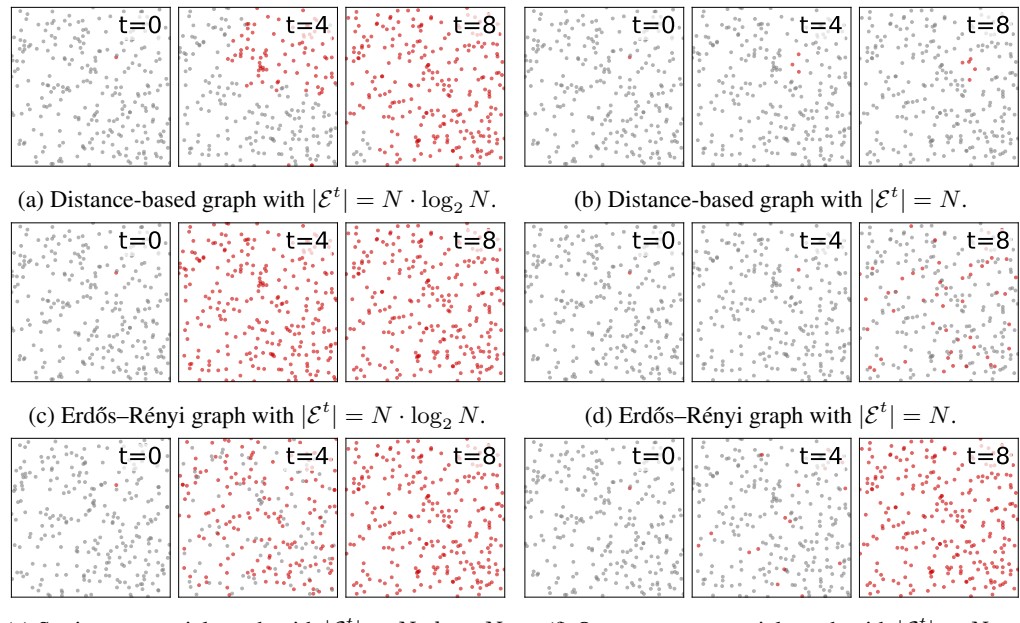

(a) Distance-based graph with $|\mathcal{E}^t| = N \cdot \log_2 N$.    (b) Distance-based graph with $|\mathcal{E}^t| = N$.

(c) Erdős–Rényi graph with $|\mathcal{E}^t| = N \cdot \log_2 N$.    (d) Erdős–Rényi graph with $|\mathcal{E}^t| = N$.

(e) Static exponential graph with $|\mathcal{E}^t| = N \cdot \log_2 N$.    (f) One-peer exponential graph with $|\mathcal{E}^t| = N$.

Figure 2: A toy example to illustrate the message dissemination with different graph topologies. We demonstrate how the messages, represented by red dots, travel from a random agent to other agents over time, following different graph structures. In distance-based graphs (Jiang et al., 2020), agents are connected to top-$K$ nearest neighbors. In Erdős–Rényi random graphs (Erdos et al., 1960), the adjacency matrices are sampled uniformly from all the graphs satisfying the diameter and size conditions. In exponential graphs, the adjacency matrices follow Equations (1) and (2).

**One-peer Exponential Graph**    In the one-peer exponential graph, each agent iterates through different peers that are $2^0, 2^1, \ldots, 2^{\lfloor \log_2 (N-1) \rfloor}$ hops away, which is illustrated by Figure 1b. Formally, we have

$$E_{ij}^{t\,(\text{one-peer})} = \begin{cases} 1 & \text{if } \log_2\left((j-i) \bmod N\right) = t \bmod \lfloor \log_2(N-1) \rfloor \text{ or } i = j \\ 0 & \text{otherwise} \end{cases}. \tag{2}$$

**Properties**    Using the adjacency matrices defined above, we verify that the graph diameter for both static and one-peer exponential graphs is $\lceil \log_2(N-1) \rceil$ (see Appendix A for details). As discussed in Section 3.1.2, a small diameter facilitates efficient information dissemination, especially when the number of agents $N$ is large.

Regarding communication costs, static exponential graphs have a size of $N \cdot \lfloor \log_2(N-1) \rfloor$, while one-peer exponential graphs have a size of $N$. Notably, the size of one-peer exponential graphs scales linearly with the number of agents, meaning the overall communication overhead also scales linearly.

To illustrate these properties, we provide a toy example in Figure 2. We visualize the message dissemination abilities of different communication topologies under varying communication budgets. In this example, with $N = 256$ agents, graph sizes (communication budgets) $|\mathcal{E}^t|$ are set to $N \cdot \log_2 N$ and $N$, respectively. In Figure 2, we observe that for each communication topology, reducing the graph sizes (as shown in Figures 2b, 2d and 2f) typically slows down dissemination speed due to increased graph diameters. This illustrates a trade-off between graph diameter and size, reflecting the trade-off between communication performance and overhead in many-agent systems. Sparser graphs with smaller sizes result in slower message dissemination but lighter communication overhead. However, exponential topologies strike a balance in this trade-off, demonstrating strong information diffusion even with a minimal communication budget of $N$.

Based on these observations, we conclude that exponential topologies are well-suited for many-agent communication because: 1) In exponential topologies, any two agents can exchange messages in

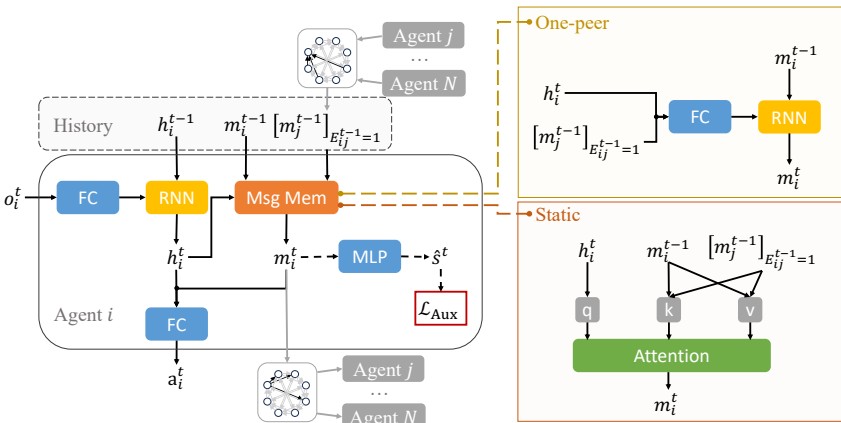

Figure 3: Neural network architecture for ExpoComm. For the static exponential topologies, attention blocks are used for message aggregation. For the one-peer exponential topologies, RNN blocks are used for message aggregation.

at most $\lceil \log_2 (N-1) \rceil$ timesteps, ensuring timely information exchange in decentralized decision-making problems. 2) The communication overhead scales nearly linearly with the number of agents, which is crucial for many-agent systems. 3) With a rule-based topology, exponential graphs are easy to deploy and adapt to systems with varying numbers of agents, as empirically verified in Section 4.2.

## 3.2 NEURAL NETWORK ARCHITECTURE DESIGN

With exponential graphs serving as the communication topology in ExpoComm, we we elaborate on the neural network architecture to help agents utilize received messages for better decision-making. The overall architecture is illustrated in Figure 3. ExpoComm is based on the concept of facilitating message flow across all agents within a certain timeframe, where the graph diameter indicates the length of such timeframe. To capitalize on the small graph diameter of exponential graphs, the message-processing module at each agent should ideally preserve all information received within diameter($\mathcal{G}^t$) timesteps. However, preserving all messages across multiple timesteps is not memory-efficient, so we employ sequential neural networks, such as attention blocks and recurrent neural networks (RNNs), for message processing.

## 3.3 TRAINING AND EXECUTION DETAILS

Following the QMIX (Rashid et al., 2020) algorithm, we update the network parameters $\theta$ with the objective of minimizing the temporal difference (TD) error loss:

$$\mathcal{L}^{\text{TD}}(\theta) = \mathbb{E}_{(s^t, \boldsymbol{o}^t, \boldsymbol{a}^t, r^t, s^{t+1}, \boldsymbol{o}^{t+1}) \sim \mathcal{D}} \left[ \left( y^{tot} - Q_{tot}(s^t, \boldsymbol{o}^t, \boldsymbol{a}^t; \theta) \right)^2 \right], \tag{3}$$

where $y^{tot} = r + \gamma \max_{\boldsymbol{a}} Q_{tot}(s^{t+1}, \boldsymbol{o}^{t+1}, \boldsymbol{a}; \theta^-)$ and $\theta^-$ represents the parameters of the target network as in DQN.

However, communication inevitably enlarges the policy space, making it more challenging to find the optimal policy relying solely on the MARL training objective (Li & Zhang, 2024). To facilitate learning meaningful messages, we introduce auxiliary tasks to restore global information from local messages. From a message perspective, we aim for it to traverse among agents over multiple timesteps, accumulating new information along the way, and ultimately reflecting global information useful for decision-making.

**Message grounding with the global state** In scenarios where the global state is available during training, the auxiliary loss is given by the prediction error of the current global state:

$$\mathcal{L}_{\text{pred}}^{\text{Aux}}(\theta, \phi) = \mathbb{E}_{(s^t, \boldsymbol{o}^t) \sim \mathcal{D}} \left[ s^t - f(m_i^t; \phi))^2 \right], \tag{4}$$

where the learnable auxiliary network for prediction $f(\cdot; \phi)$ is used to ground the messages and can be discarded after training.

---

**Algorithm 1** Training and Execution Procedure of ExpoComm

---

1: **Init:** Network parameters $\theta$, $\phi$, $\mathcal{D} = \emptyset$, step $= 0$, $\theta^- = \theta$
2: **while** step $<$ step$_{\text{max}}$ **do**
3:     $t = 0$. Reset the environment.
4:     **for** $t = 1, 2, ...,$ episode_limit **do**
5:         *// Decentralized execution at agent $i$*
6:         Update local history $h_i^t$ based on current observation $o_i^t$ and previous history $h_i^{t-1}$
7:         Update agent $i$'s message $m_i^t$ based on previous local message $m_i^{t-1}$ and previously received messages $\left[ m_j^{t-1} \right]_{E_{ij}^{t-1}=1}$
8:         *// Communication*
9:         Send message $m_i^t$ to peers $\{j \mid E_{ij}^t = 1\}$               $\{\triangleright$ Equations (1) and (2)$\}$
10:       *// Action, which can happen concurrently with communication*
11:       Sample action $a_i^t$ based on current history $h_i^t$ and current message $m_i^t$
12:       Interact with the environment $(s^{t+1}, \boldsymbol{o}^{t+1}, r^t) = \text{env.step}(\boldsymbol{a}^t)$
13:       Save the experience $\mathcal{D} = \mathcal{D} \cup \left( s^t, \boldsymbol{o}^t, \boldsymbol{a}^t, r^t, s^{t+1}, \boldsymbol{o}^{t+1} \right)$
14:     **end for**
15:     At some interval, update network parameters $\theta$, $\phi$ and $\theta^-$         $\{\triangleright$ Equation (6)$\}$
16: **end while**
17: **Output:** Policy networks parameters $\theta$

---

**Message grounding without the global state**   Alternatively, when the global state is unavailable during training, we use contrastive learning for meaningful message encoding, similar to Lo et al. (2024). Specifically, we treat messages from different agents at the same timestep as positive pairs and messages with intervals larger than diameter($\mathcal{G}^t$) as negative pairs, encouraging local messages $m_i^t$ to reflect the current global latent state. The corresponding auxiliary loss is given as the InfoNCE loss (Oord et al., 2018):

$$\mathcal{L}_{\text{cont}}^{\text{Aux}}(\theta) = -\mathbb{E}_{i,j,t,t'} \left[ \log \frac{\exp \left( g(m_i^t) \cdot g(m_j^t)/\tau \right)}{\sum_{m \in \mathcal{M}} \exp \left( g(m_i^t) \cdot g(m)/\tau \right)} \right], \tag{5}$$

where $i$ is uniformly sampled from $\{0, \ldots, N\}$, $j$ is uniformly sampled from $\{0, \ldots, N : j \neq i\}$, $\mathcal{M} = \{m_k^{t'} : k \in \{0, \ldots, N\}, t' \notin [t - \text{diameter}(\mathcal{G}^t), t + \text{diameter}(\mathcal{G}^t)]\} \cup \{m_j^t\}$ with $|\mathcal{M}| = M + 1$ and $m$ is uniformly sampled from $\mathcal{M}$. $g(\cdot)$ is the normalization function, $M$ is the hyperparameter indicating the number of negative pairs and $\tau$ is the temperature hyperparameter. The overall training loss is:

$$\mathcal{L}^{\text{TD}}(\theta) = \mathcal{L}^{\text{TD}}(\theta) + \alpha \cdot \mathcal{L}^{\text{Aux}}(\theta; \phi), \tag{6}$$

where $\alpha$ is the hyperparameter and $\mathcal{L}^{\text{Aux}}(\cdot)$ is the auxiliary loss defined by Equation (4) or Equation (5), depending on whether global information is available during training. The training and execution procedures are summarized in Algorithm 1.

## 4   EXPERIMENTAL RESULTS

In this section, we evaluate ExpoComm on two large-scale multi-agent benchmarks: MAgent (Zheng et al., 2018) and Infrastructure Management Planning (IMP) (Leroy et al., 2024). All experiments are averaged over five random seeds and the shaded areas represent the 95% confidence interval. Details on network architecture and the training hyperparameters are available in Appendix B.1.

### 4.1   EXPERIMENTAL SETUPS

**Environment descriptions**   In this section, We test ExpoComm and baselines across twelve scenarios in two large-scale benchmarks, with the number of agents ranging from 20 to 100. Specifically, MAgent is a particle-based gridworld environment representative of the typical MARL gaming benchmarks. To expand the variety of tasks, we also include the IMP benchmark, with tasks oriented from real-world applications. More details regarding the environment settings are provided in Appendix B.2.

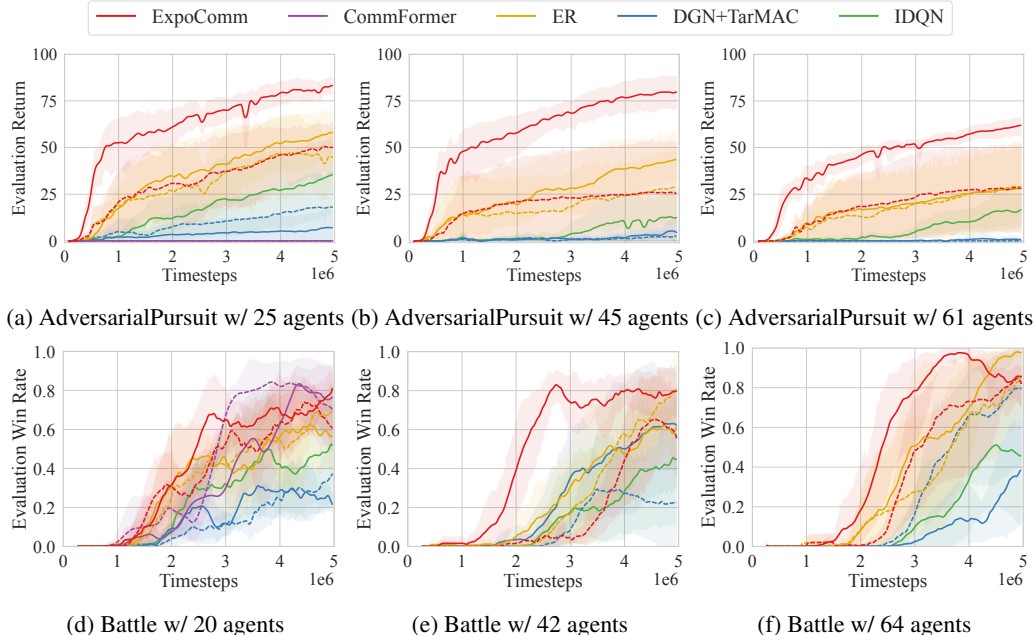

Figure 4: Performance comparison with baselines on MAgent tasks. Solid lines represent communication budgets of $K = 1$, while dashed lines represent budgets of $K = \lceil \log_2 N \rceil$. Runs requiring more than 40 GB of GPU memory are excluded due to extreme training costs compared to other methods.

**Communication Budgets** Denoting the number of agents each agent communicates to by $K$, for each baseline in each scenario, we test two communication budgets: $K = \lceil \log_2 N \rceil$ and $K = 1$, where $N$ is the number of agents in the systems.

**Baselines** In the following, we compare our proposed ExpoComm with four baselines: (i) *IDQN/QMIX* (Rashid et al., 2020): Base algorithms without communication; (ii) *DGN+TarMAC* (Jiang et al., 2020; Das et al., 2019): Position-based communication topologies in which agents communicate with their nearest neighbors and use TarMAC structure to aggregate messages; (iii) *ER*: ExpoComm with the exponential graph topologies replaced by random graph communication topologies following the Erdős–Rényi model; (iv) *CommFormer* (Hu et al., 2024): Learned communication topologies using GNN. For ExpoComm, we use the static exponential graph variant for $K = \lceil \log_2 N \rceil$ and the one-peer exponential graph variant for $K = 1$. For DGN+TarMAC, agents communicate to top-$K$ nearest neighbors. For ER, communication graphs are sampled uniformly from all the $K$-in-regular directed graphs. CommFormer uses constraints with varying sparsity levels for different communication budgets. Official implementations of these baselines are utilized wherever available; otherwise, we closely follow the descriptions from their respective papers, integrating them into the base algorithms. More implementation details can be found in Appendix B.3.

## 4.2 RESULTS

**Benchmark results** We present the comparative performance of ExpoComm and baselines in MAgent and IMP environments with Figure 4 and Table 1, respectively. Overall, ExpoComm demonstrates superior performance in these large-scale benchmarks under both communication budgets, underscoring the scalability and robustness of ExpoComm strategies. Notably, the one-peer version of ExpoComm achieves the best performance in most scenarios, despite communication costs that only grow linearly with the number of agents. This makes it the most suitable method for handling large-scale MARL communication problems under very low communication budgets. Additional visualization results to illustrate the learned policies are provided in Appendix C.1.

**Zero-shot transfer** Similar to the experimental settings suggested by Wang et al. (2022), we test the zero-shot transfer ability of our proposed ExpoComm and the baseline methods, reporting the

Table 1: Performance comparison with baselines on IMP tasks. Results are reported as the mean and standard deviation of the percentage of normalized discounted rewards relative to expert-based heuristic policies, following Leroy et al. (2024), with details in Appendix B.2. The best-performing method is indicated in **bold**, and the second best is underlined.

| Scenario | QMIX | ER | | ExpoComm | |
|---|---|---|---|---|---|
| | $K = 0$ | $K = 1$ | $K = \lceil \log_2 N \rceil$ | $K = 1$ | $K = \lceil \log_2 N \rceil$ |
| | | | $N = 50$ | | |
| Uncorrelated | 26.42 (3.43) | 24.91 (3.77) | 26.62 (2.03) | 27.31 (2.26) | **28.26** (2.51) |
| Correlated | 24.81 (4.16) | 34.63 (9.72) | 34.76 (5.07) | **43.82** (6.33) | 40.01 (3.19) |
| OWF | 62.45 (3.46) | 62.99 (3.02) | 61.70 (4.62) | 64.66 (0.26) | **65.19** (0.51) |
| | | | $N = 100$ | | |
| Uncorrelated | 12.86 (6.88) | 21.94 (5.97) | 18.36 (12.92) | 27.34 (13.32) | **27.81** (5.71) |
| Correlated | $-40.20$ (96.35) | $-65.14$ (65.08) | 9.84 (32.27) | **19.17** (23.94) | 17.25 (22.70) |
| OWF | 65.55 (0.53) | **66.70** (0.50) | 65.92 (0.87) | 65.26 (1.34) | 66.23 (0.38) |

[1] DGN+TarMAC is not suitable for this benchmark because it requires the physical positions of agents, which are not available in this environment.

[2] Methods that require more than 40 GB GPU memory are excluded from comparison due to the extreme training cost compared to other methods.

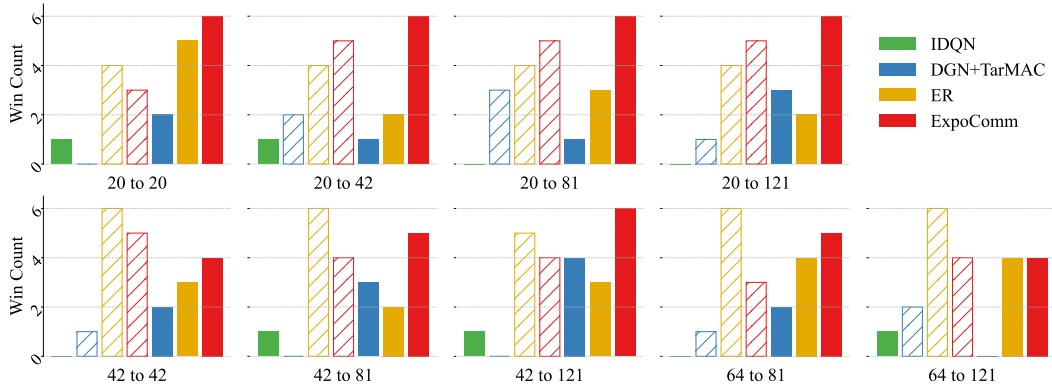

Figure 5: Zero-shot transfer results on `Battle` scenario. The subtitle "X to Y" indicates that methods are trained with X agents and tested with Y agents. Filled bars represent communication budgets of $K = \lceil \log_2 N \rceil$, while hatched bars represent budgets of $K = 1$. Baseline CommFormer is not excluded in this experiment because it learns a fixed peer-to-peer communication topology among agents in a specific scenario and it is non-trivial to transfer such topology to scenarios with different numbers of agents.

results in Figure 5. Specifically, we train the agent policies and their corresponding communication policies in scenarios with smaller numbers of agents and directly test these policies against each other in larger agent scenarios in the competitive task `Battle`. We test each pair of methods over 200 games, record the method with more wins as the winner, and summarize the results in Figure 5. We observe that both ER and ExpoComm demonstrate good transfer ability compared to other baselines, with ExpoComm performing better under smaller communication budgets. The superior transfer ability of ER and ExpoComm may be attributed to the grounding of messages, which reflects global information.

**Ablation studies** We conduct ablation studies to assess the impact of various design elements in ExpoComm, with results presented in Figure 6. In particular, we compare ExpoComm with two ablations: (i) *ExpoComm w/o mem*, in which the message generators are not memory-based as described in Section 3.2. (ii) *ExpoComm w/o aux*, which lacks the auxiliary loss term described in Section 3.3. From the results, we see that removing the memory blocks from the message generators hinders effective message generation, especially in scenarios with strong time correlation such as

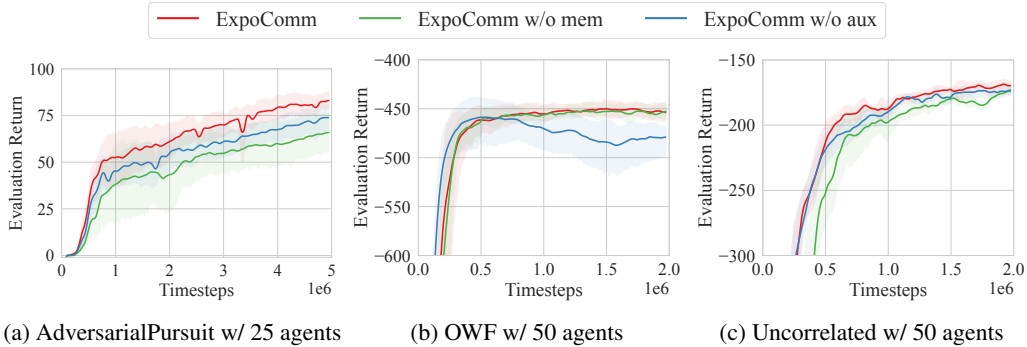

(a) AdversarialPursuit w/ 25 agents    (b) OWF w/ 50 agents    (c) Uncorrelated w/ 50 agents

Figure 6: Ablation studies on MAgent and IMP benchmarks.

`AdversarialPursuit`. Auxiliary tasks primarily aid in grounding the messages, without which the messages could lack guidance and even be detrimental to decision-making.

**Discussion** As discussed in Section 2, existing methods suitable for large-scale multi-agent communication fall into two categories: position-based methods (e.g., DGN+TarMAC) and GNN-based methods (e.g., CommFormer). We propose a third category based on specific graph structures. In this category, ExpoComm utilizes exponential topologies, and we construct the baseline ER using Erdős–Rényi random graphs. Based on our analysis and experiments, the advantages of ExpoComm are as follows:

- **Superior task performance**: Due to the small diameter of exponential graphs and memory-based message generators, ExpoComm facilitates fast message dissemination among all agents. It efficiently collects and carries local information from all agents, aiding decentralized decision-making. This advantage is supported by experiments shown in Figure 4 and Table 1.

- **Low communication costs**: The compact size of exponential graphs ensures that ExpoComm's communication costs scale (near-)linearly with the number of agents $N$, crucial for managing communication costs in multi-agent systems. Unlike position-based methods or ER, which can only control the number of in-edges or out-edges without global scheduling, ExpoComm naturally balances communication overhead across agents.

- **Versatile adaptability**: ExpoComm shows strong transferability across different numbers of agents, as seen in Figure 5. This is due to its global message dissemination strategies, which focus on overall communication rather than pairwise relationships, allowing it to adapt to more agents. Additionally, ExpoComm handles a wide range of tasks, regardless of the task nature or agent count. Unlike position-based methods, which may struggle with non-gridworld tasks like IMP due to assumptions about knowledge of agent locations, ExpoComm makes no such assumptions. Moreover, while learning effective pairwise communication topologies using GNNs can lead to significant GPU memory consumption, ExpoComm bypasses these challenges. It does not rely on the expensive task of learning a scenario-specific communication topology guided by the MARL task itself but instead uses a well-designed rule-based topology based on the communication desiderata analyzed in Section 3.1.2.

## 5 CONCLUSIONS

In this work, we explored scalable communication strategies in MARL and introduced ExpoComm, an exponential topology-enabled communication protocol. We proposed a framework with communication topologies featuring small diameters for fast information dissemination and small graph sizes for low communication overhead. This framework is complemented by memory-based message processors and message grounding through auxiliary objectives to achieve effective global information representation. Despite requiring only (near-)linear communication costs relative to the number of agents, ExpoComm demonstrated superior performance and strong transferability on large-scale benchmarks like MAgent and IMP. This study highlights the potential for enhancing the scalability of MARL communication strategies through the explicit design of communication topologies.

## REPRODUCIBILITY STATEMENT

Method and implementation details are provided in Section 3, Appendix B.1, and Appendix B.3. Experiment settings and details are described in Section 4.1 and Appendix B.2. Information about the experimental infrastructure is available in Appendix B.4. The code is publicly available at `https://github.com/LXXXXR/ExpoComm`.

## ACKNOWLEDGMENT

This work was supported by the Hong Kong Research Grants Council under the NSFC/RGC Collaborative Research Scheme grant CRS_HKUST603/22 and the Shanghai Sailing Program 24YF2710200. We thank the anonymous reviewers for their valuable feedback and suggestions.

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

## A  THEORETICAL ANALYSIS

In this section, we analyze the communication effect of exponential topologies.

**Theorem 1.** *Suppose that $E_{ij}^t$ is defined by Equation* (2). *Let $\tau = \lceil \log_2 (N - 1) \rceil$. Then, the following holds:*

$$E_{ij}^t \times_b E_{ij}^{t+1} \times_b \dots E_{ij}^{t+\tau-1} = \mathbb{1}\mathbb{1}^T, \tag{7}$$

*where $\times_b$ denotes logical (Boolean) matrix multiplication.*

**Remark 1.** *If the information at each agent remains valid within $\tau$ timesteps and there is no information loss during aggregation, the one-peer exponential topology ensures information exchange between any two agents in the system with $\tau$ timesteps.*

**Remark 2.** *Static exponential topologies exhibit a similar communication effect as described in Theorem 1. Specifically, $\forall i, j$ that $E_{ij}^{t(one\text{-}peer)} = 1$, it holds that $E_{ij}^{t(stat)} = 1$.*

*Proof.* Define function $Z : \mathbb{R}_+ \to \{0, 1\}$ such that

$$Z(x) = \begin{cases} 1, & x > 0, \\ 0, & x = 0. \end{cases} \tag{8}$$

Then, for all $x, y, u, v \geq 0$, the following equivalence holds:

$$xy + wv = 0 \iff (Z(x) \times_b Z(y)) +_b (Z(u) \times_b Z(v)) = 0, \tag{9}$$

where $\times_b$ denotes logical (Boolean) `And`, and $+_b$ denotes logical (Boolean) `Or`. Now, consider the connection between $Z(x)$ and the structure of an all-one matrix. For a non-negative matrix $X$, it holds that $X_{ij} \in \mathbb{R}^+, \forall i, j \iff Z(X) = \mathbb{1}\mathbb{1}^T$.

Therefore, by applying Appendix A to $E_{ij}^t E_{ij}^{t+1} \dots E_{ij}^{t+\tau-1} = \frac{2^\tau}{N} \mathbb{1}\mathbb{1}^T$ (Ying et al., 2021), we have $E_{ij}^t \times_b E_{ij}^{t+1} \times_b \dots E_{ij}^{t+\tau-1} = \mathbb{1}\mathbb{1}^T$. $\qquad\square$

## B  EXPERIMENT DETAILS

### B.1  NETWORK ARCHITECTURE AND HYPERPARAMETERS

**Codebase**    Our implementation of ExpoComm and baseline algorithms is based on the following codebase:

- EPyMARL (Papoudakis et al., 2021): `https://github.com/uoe-agents/epymarl`
- CommFormer Hu et al. (2024): `https://github.com/charleshsc/CommFormer`
- CommNet (Sukhbaatar et al., 2016): `https://github.com/isp1tze/MAProj`

The code for ExpoComm is publicly available at `https://github.com/LXXXXR/ExpoComm`.

**Neural network architecture**    Following previous work Papoudakis et al. (2021), we employ deep neural networks consisting of multilayer perceptrons (MLPs) with rectified linear unit (ReLU) activation functions and gated recurrent units (GRUs) to parameterize the agent networks. In ExpoComm, the message memory blocks described in Section 3.2 are implemented using a single GRU or an attention block. The prediction network $f(\cdot; \phi)$ described in Section 3.3 is implemented using a two-layer MLP.

**Hyperparameters**    To ensure a fair comparison, we implement our method and self-constructed baselines using the same codebase with the same set of hyperparameters, with the exception of method-specific ones and the learning rate. In general, we follow the common settings provided by Papoudakis et al. (2021) for MAgent benchmark and adopt the settings in IMP paper (Leroy et al., 2024) for the IMP benchmark. The common hyperparameters are listed in Table 2. The ExpoComm-specific hyperparameters are provided in Table 3. For learning rate, we search among

$(0.0001, 0.0005)$ for ExpoComm and baselines. We use the value of $0.0005$ for base algorithms without communication; $0.0005$ for DGN+TarMAC in MAgent and $0.0001$ for DGN+TarMAC in IMP; $0.0001$ for ExpoComm in IMP with 50 agents and $0.0005$ for other scenarios. For Comm-Former, we adopt the optimal value in its official implementation.

Table 2: Common hyperparameters.

| Hyperparameter | Benchmark | Value |
|---|---|---|
| Hidden sizes | - | 64 |
| Discount factor $\gamma$ | MAgent | 0.99 |
| | IMP | 0.95 |
| Batch size | MAgent | 32 |
| | IMP | 64 |
| Replay buffer size | - | 2000 |
| Number of environment steps | MAgent | $5 \times 10^6$ |
| | IMP | $2 \times 10^6$ |
| Epsilon anneal steps | MAgent | $5 \times 10^5$ |
| | IMP | $5 \times 10^3$ |
| Test interval steps | MAgent | $5 \times 10^4$ |
| | IMP | $2.5 \times 10^4$ |
| Number of test episode | - | 100 |

Table 3: Hyperparameters used for ExpoComm.

| Hyperparameter | Value |
|---|---|
| Auxillary loss coefficient $\alpha$ | 0.1 |
| Temperature $\tau$ | 0.07 |
| Number of negative pairs $M$ | 20 |

## B.2 Environmental details

**Codebase**  The environments used in this work are listed below with descriptions in Table 4.

- MAgent (Zheng et al., 2018; Terry et al., 2020):
  `https://github.com/Farama-Foundation/MAgent2`
- IMP (Leroy et al., 2024): `https://github.com/moratodpg/imp_marl`

Table 4: Environments details.

| Environment | Scenarios | Number of agents |
|---|---|---|
| MAgent | Adversarial Pursuit | $(25, 45, 61)$ [1] |
| | Battle | $(20, 42, 64)$ [2] |
| IMP | Uncorrelated: uncorrelated k-out-of-n; campaign cost | $(50, 100)$ |
| | Correlated: correlated k-out-of-n; campaign cost | $(50, 100)$ |
| | OWF: offshore wind farm; campaign cost | $(50, 100)$ |

[1] The number of agents in this scenario is determined by setting the map size to $25, 35, 40$, respectively.

[2] The number of agents in this scenario is determined by setting the map size to $45, 60, 70$, respectively.

**MAgent**  MAgent is a highly scalable gridworld gaming benchmark shown in Figure 7. In `AdversarialPursuit`, agents aim to tag adversaries while adversaries try to escape. Agents

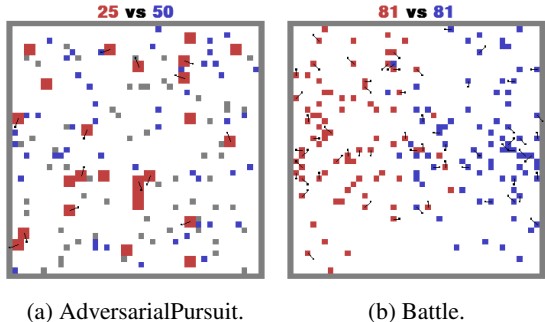

(a) AdversarialPursuit.                    (b) Battle.

Figure 7: Environments from the MAgent benchmark suite (Terry et al., 2020). In each scenario, the MARL algorithms control the red agents, while the blue adversary agents are controlled by pretrained policies.

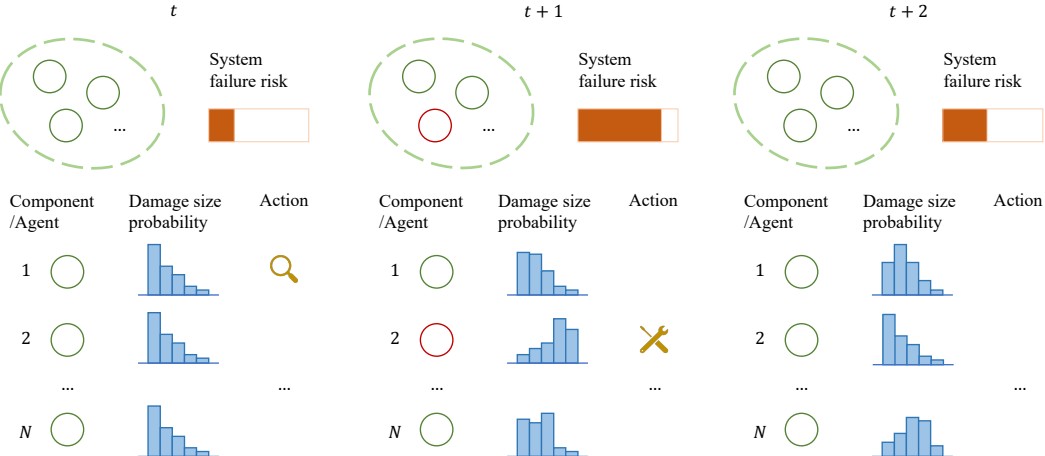

Figure 8: IMP environment (Leroy et al., 2024). This environment simulates an engineering system with multiple components controlled by agents. The objective is to minimize overall system failure risk at low costs. The system risk depends on component damage probabilities, which evolve over time and can be influenced by agent inspection or repair actions.

can choose actions from `move`, `tag` or `do nothing`. Agents are rewarded for successfully tagging an adversary and penalized for unsuccessful tagging attempts. In `Battle`, agents attempt to attack and eliminate adversaries, with the same goal for adversaries. Agents can choose actions from `move`, `attack` or `do nothing`. A team wins by eliminating all opponents or having more surviving agents when the episode ends.

Following the official implementation (Terry et al., 2020), we use an individual reward setting with IDQN as the base algorithm. Due to the high-dimensional observations in MAgent, storing experience in a replay buffer can be challenging because of hardware constraints. We adopt a preprocessing procedure following Jiang et al. (2020), compressing observations by concatenating [my_team_hp - obstacle/off the map, other_team_hp - obstacle/off the map]. To facilitate the use of communication, we use small view ranges (8 for `AdversarialPursuit` and 7 for `Battle`). In both scenarios, we pretrain the adversary policies using the IDQN algorithm with a self-play scheme and use these pretrained policies to test the performance of different algorithms.

**IMP**    IMP is a platform for benchmarking the scalability of cooperative MARL methods in real-world engineering applications, as illustrated in Figure 8. This environment simulates an infrastructure management planning problem with agents controlling different components. Agents can choose actions from `inspection`, `repair` or `do nothing`. In different scenarios, the correlation between agent deterioration processes and the system failure function are defined differently, posing unique challenges.

Following the official implementation of IMP, we use a global reward setting and choose QMIX as the base algorithm due to its stable performance across scenarios. We adopt the campaign cost setting, which requires higher cooperation among agents. As recommended by Leroy et al. (2024), results are normalized with respect to expert-based heuristic policies using $(x - H)/|H|$, where $x$ is the discounted rewards of the tested algorithm, and $H$ is the discounted rewards achieved by heuristic policies listed in Table 5.

Table 5: Heuristic policies performance on the IMP benchmark.

| Scenario | Number of agents $N$ | Discounted reward $H$ |
|---|---|---|
| Uncorrelated | 50 | $-232.7$ |
| | 100 | $-231.5$ |
| Correlated | 50 | $-211.0$ |
| | 100 | $-194.0$ |
| OWF | 50 | $-1248.2$ |
| | 100 | $-2436.3$ |

### B.3 IMPLEMENTATION DETAILS

In MAgent, we implement our proposed ExpoComm along with baselines DGN+TarMAC and ER on top of the IDQN base algorithm. In IMP, these are implemented on top of QMIX. For ExpoComm, we use Equation (4) for MAgent benchmark because the global state is provided in this environment, and Equation (5) for IMP, as the global state is a concatenation of all observations and is not compact or suitable for message grounding.

### B.4 EXPERIMENTAL INFRASTRUCTURE

The experiments were conducted using NVIDIA GeForce RTX 3080 GPUs and NVIDIA A100GPUs. Each experimental run required less than 2 days to complete.

## C MORE RESULTS AND DISCUSSION

### C.1 VISUALIZATION RESULTS

We visualize the final trained policies of ExpoComm and IDQN in `dversarialPursuit` and `Battle` with Figure 9 and Figure 10 respectively to demonstrate how ExpoComm enhances cooperation among agents. As shown in Figure 9a and Figure 10a, agents adopt a global perspective and act cooperatively with ExpoComm policies, demonstrating effectiveness even under extreme communication budgets($K = 1$). In comparison, IDQN agents focus only on local observations and often become trapped in suboptimal solutions due to lack of coordination.

### C.2 COMPARISON WITH PROXY-BASED COMMUNICATION

Although we primarily focus on decentralized communication-based MASs without centralized proxies, we also compare ExpoComm against the proxy-based CommNet (Sukhbaatar et al., 2016). As seen in Figure 11 and Table 6, ExpoComm outperforms CommNet in most scenarios, especially in IMP benchmarks. However, CommNet achieves comparable performance on the `AdversarialPursuit` tasks. This implies that a global perspective is more crucial for success in these scenarios, possibly explaining ExpoComm's larger advantage over other baselines in this scenario.

### C.3 LIMITATIONS AND FUTURE WORK

While ExpoComm demonstrates strong performance and scalability in cooperative multi-agent tasks, some limitations remain.

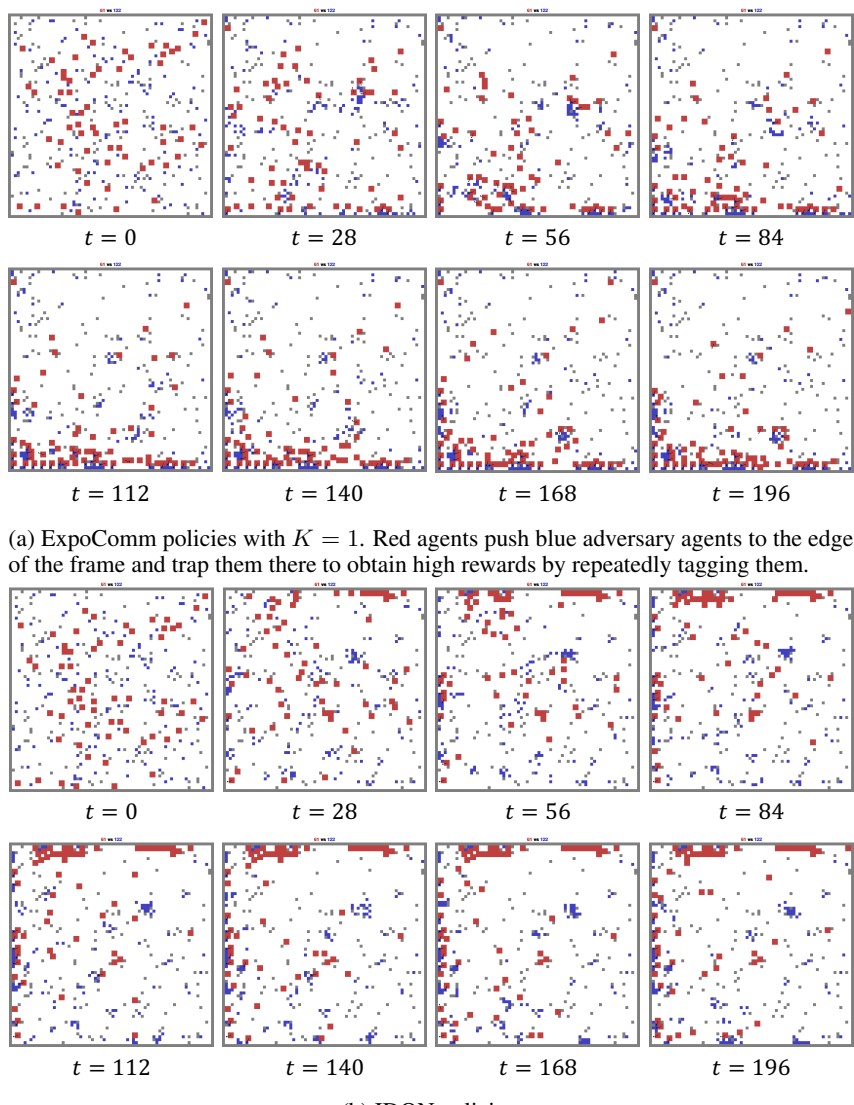

$t = 0$     $t = 28$     $t = 56$     $t = 84$

$t = 112$     $t = 140$     $t = 168$     $t = 196$

(a) ExpoComm policies with $K = 1$. Red agents push blue adversary agents to the edge of the frame and trap them there to obtain high rewards by repeatedly tagging them.

$t = 0$     $t = 28$     $t = 56$     $t = 84$

$t = 112$     $t = 140$     $t = 168$     $t = 196$

(b) IDQN policies.

Figure 9: Visualization in AdversarialPursuit w/ 61 agents.

First, ExpoComm does not explicitly incorporate agent heterogeneity or properties of the underlying environmental MDP when constructing the communication topology. This could result in suboptimal performance in scenarios requiring targeted messaging between specific agents (Yuan et al., 2022) or in networked MDPs (Zhang et al., 2018; Ma et al., 2024). Therefore, Incorporating factors like agent identities or relationships presents a promising direction for further improvements in such settings.

Second, we evaluated ExpoComm primarily in fully cooperative tasks. Partially competitive settings requiring agents to learn to communicate only when necessary remain challenging. Examining ExpoComm's capabilities and limitations in such partially competitive tasks presents an important avenue for future work.

Finally, communication scalability in multi-agent systems remains an under-explored area despite the attempt of this work. For instance, incorporating finer graph topologies beyond exponential graphs may enhance performance, and exploiting temporal communication sparsity could further reduce costs. There are still many open questions in scaling communication efficiently.

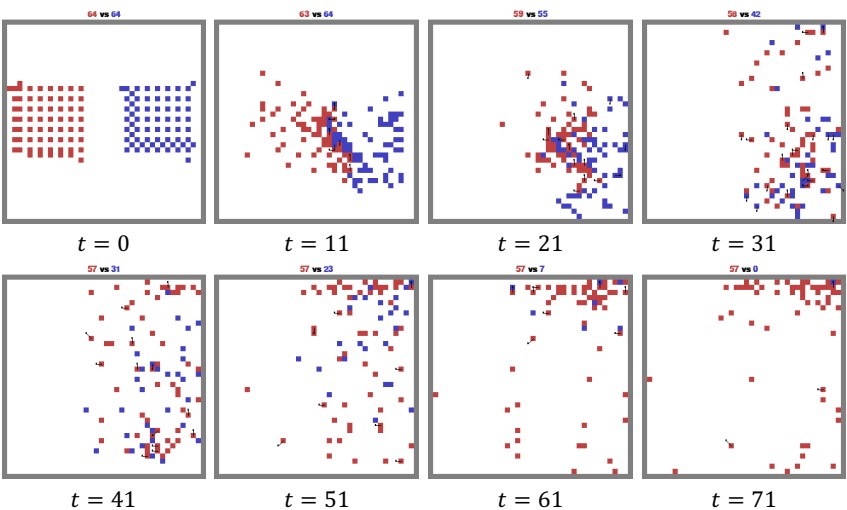

(a) ExpoComm policies with $K = 1$. Agents coordinate to ensure red agents outnumber blue adversaries on the front line ($t = 11$, $t = 21$), securing an advantage. Once red agents substantially outnumber blue adversaries, they surround the remaining adversaries ($t = 61$, $t = 71$) to eliminate them.

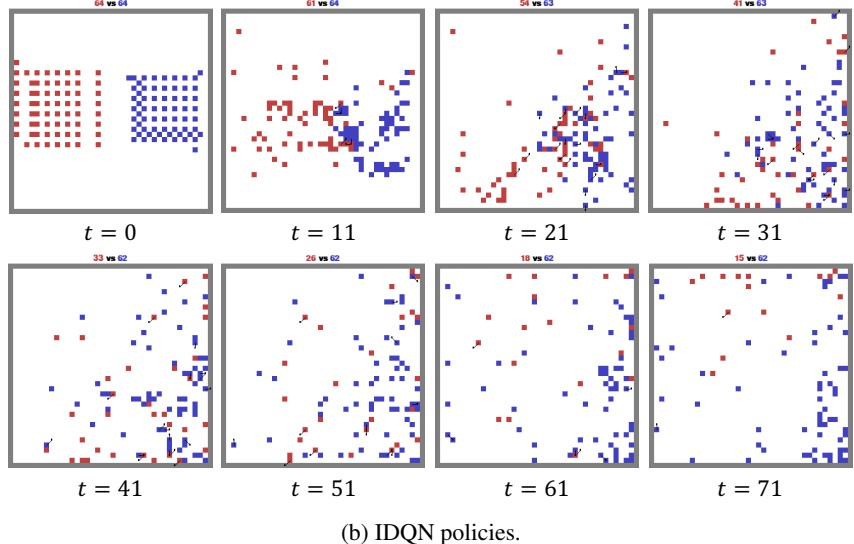

(b) IDQN policies.

Figure 10: Visualization in Battle w/ 64 agents.

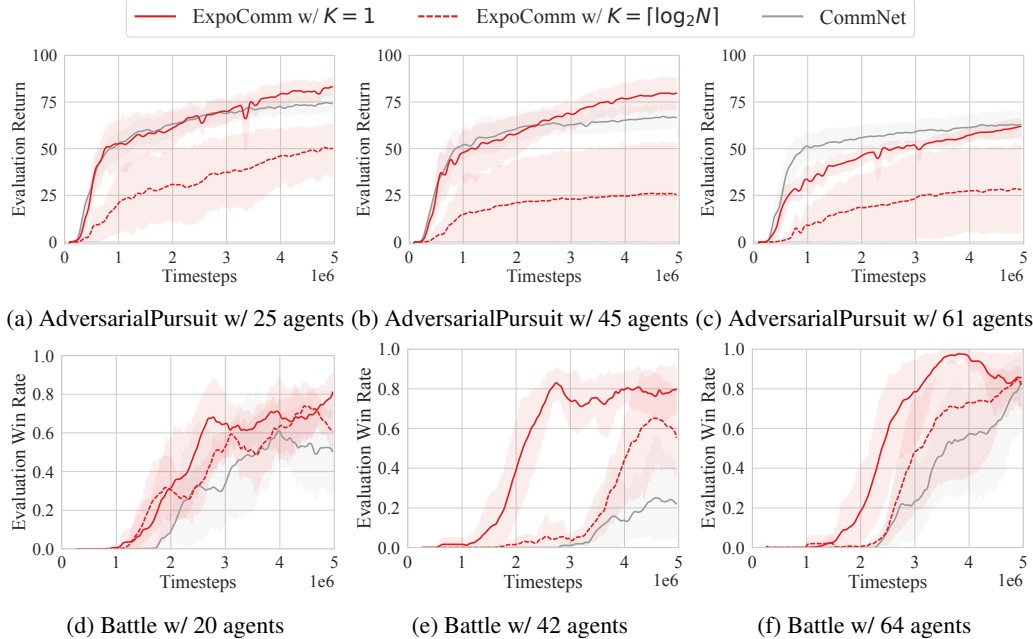

(a) AdversarialPursuit w/ 25 agents (b) AdversarialPursuit w/ 45 agents (c) AdversarialPursuit w/ 61 agents

(d) Battle w/ 20 agents       (e) Battle w/ 42 agents       (f) Battle w/ 64 agents

Figure 11: Performance comparison with proxy-based baselines on MAgent tasks.

Table 6: Performance comparison with proxy-based baselines on IMP tasks. Results are reported as the mean and standard deviation of the percentage of normalized discounted rewards relative to expert-based heuristic policies, following Leroy et al. (2024), with details in Appendix B.2. The best-performing method is indicated in **bold**, and the second best is underlined.

| Scenario | CommNet | ExpoComm | |
| --- | --- | --- | --- |
| | with communication proxy | $K = 1$ | $K = \lceil \log_2 N \rceil$ |
| $N = 50$ | | | |
| Uncorrelated | 26.07 (6.82) | 27.31 (2.26) | **28.26** (2.51) |
| Correlated | 26.14 (16.87) | **43.82** (6.33) | 40.01 (3.19) |
| OWF | 53.71 (1.27) | 64.66 (0.26) | **65.19** (0.51) |
| $N = 100$ | | | |
| Uncorrelated | −65.92 (125.03) | 27.34 (13.32) | **27.81** (5.71) |
| Correlated | −82.76 (48.62) | **19.17** (23.94) | 17.25 (22.70) |
| OWF | 34.71 (5.34) | 65.26 (1.34) | **66.23** (0.38) |

