# OpenReview forum: "Exponential Topology-enabled Scalable Communication in Multi-agent Reinforcement Learning"
_ICLR.cc/2025/Conference — ICLR 2025 Poster_

### Official Review · Reviewer_6QM3 · 2024-10-30

**Soundness:** 2
**Presentation:** 3
**Contribution:** 3
**Rating:** 6
**Confidence:** 4

**Summary:**

This work focuses on the problem of learning scalable communication in many-agent systems through multi-agent reinforcement learning. To tackle this problem, this work uses exponential graphs to model the communication topology, and memory-based message processors and message grounding for information representation. Through examples and baseline comparisons, the authors demonstrate that exponential graphs can balance the trade-off between dissemination speed and redundancy; thereby allowing agents to receive messages from other connected members in the graph without additional communication overhead.

**Strengths:**

The problem this paper seeks to address is an interesting and important issue in multi-agent reinforcement learning regarding scalable communication. The idea of using exponential graphs in this application is novel to me, and I appreciated the way that the authors helped the reader build intuition for the benefits ofgraph design in section 3.1; an exponential  I thought this was clear and very well-written. The experiments on the transferability of the method were very thorough, and I appreciated the comparisons for both K cases. The ablation studies served to reiterate the author's point and useful aspects of their architecture.

**Weaknesses:**

Overall, my concerns with this paper lie in the framing regarding communication overhead.
- The definition of communication overhead in this paper is rather ambiguous. It would probably be best to add a formalized definition, as communication overhead has very field-specific connotations beyond computer science. While addressed by the authors in the introduction, it is possible to reduce communication overhead by only communicating with a subset of peers. If ExpoComm does produce lower communication overhead, I think this paper could be strengthened significantly by adding additional metrics where the number of messages (or unique messages) ExpoComm has sent is directly compared against another method that only communicates with a subset of agents (eg DGN). From my understanding, Table 1 lacks these comparisons now. If ExpoComm does not result in lower numbers of messages passed between agents against comparable baselines (eg. MAGIC, DGN), then I would recommend clarifying the wording throughout the paper, to be in line with your definition of communication overhead.

**Questions:**

-  Do you have any theories why CommFormer performs so terribly in the Adversarial Pursuit 25 agent case? It performs similarly to ExpoComm in the Battle Environment 20 agent case; I would be interested to hear your theories on why it doesn’t generalize well or performs similarly to ExpoComm in that one case.

- The performance of ER and ExpoComm are very smilar in the filled bar transferability cases (i.e. K=log2N). Do you have theories about why this is?

- Did you experiment with more K values beyond log2N and 1? If so, what did you observe and why did you believe these two values in the paper would be representative examples?

- Does communication overhead in this paper refer to number of messages passed? Can you provide a definition in-text? Are there additional communciation savings that can be achieved by your method beyond its exponential graph structure? (i.e. savings due to pruning or shorter timesteps or less redundant agents communicated with to compared to other techniques)

---

> ### Author Response · Authors · 2024-11-21
>
> Thank you for your positive review. We have updated the manuscript accordingly (highlighted in blue) and provide detailed clarifications below. If you have any follow-up questions or comments, please let us know, and we will be happy to discuss further.
>
> **Q1:** Communication overhead.
>
> **Q1.1:**
> > The definition of communication overhead in this paper is rather ambiguous... Does communication overhead in this paper refer to number of messages passed? Can you provide a definition in-text?
>
> **A1.1:** We apologize for the ambiguity. Yes, in our manuscript, communication overhead refers to the number of messages passed between agents. We have clarified this definition in Section 3.1.2 and reorganized the experiment setups in Section 4.1 to include a detailed explanation of the communication budget settings.
>
> **Q1.2:** Does ExpoComm result in lower numbers of messages passed between agents?
>
> **A1.2:** In our experimental setup, we maintain equal communication budgets (number of messages passed) across all methods except communication-free baselines. This design allows us to fairly evaluate the effectiveness of different communication strategies. Our results demonstrate that ExpoComm achieves superior performance while operating under the same communication constraints as baseline algorithms.
>
> **Q1.3:**
> > Are there additional communciation savings that can be achieved by your method beyond its exponential graph structure?
>
> **A1.3:** While our current implementation focuses on the exponential graph structure, there are promising avenues for additional communication savings. These include learning low-entropy messages [1] and exploiting temporal communication sparsity, which we discuss as future directions in Appendix C.3.
>
> **Q1.4:**
> > Did you experiment with more K values beyond log2N and 1? If so, what did you observe and why did you believe these two values in the paper would be representative examples?
>
> **A1.4:** Our choice of $K=\log_2N$ and $K=1$ was primarily motivated by theoretical considerations rather than experimental observations. These values represent logarithmic growth and constant complexity respectively, which are particularly relevant for scaling to large-scale multi-agent systems (MASs). Traditional approaches using fully-connected graphs ($K=N$) or sparse connections [2] ($K=\text{sparsity} \cdot N$, where $\text{sparsity}$ is a constant) result in quadratic overall communication costs, becoming impractical for large values of $N$ (e.g., $N=100$). By contrast, our approach deliberately explores sub-polynomial scaling through logarithmic and constant functions, offering more efficient alternatives for large-scale deployments.
>
>
> **Q2:**
> > Do you have any theories why CommFormer performs so terribly in the Adversarial Pursuit 25 agent case? It performs similarly to ExpoComm in the Battle Environment 20 agent case; I would be interested to hear your theories on why it doesn’t generalize well or performs similarly to ExpoComm in that one case.
>
> **A2:** Thank you for this thoughtful question. The issue might relate to the asymmetric nature of the AdversarialPursuit tasks. Agents move slower than adversaries and face penalties for failed tagging attempts, which can lead them to become inactive (choose to do nothing rather than trying to tag adversaries). Our analysis shows that CommFormer's training loss and gradients quickly stabilize at low values, indicating early overfitting [3]. This is likely because CommFormer has a larger number of parameters (3-5 times more than other methods), making it more prone to overfitting in this scenario.
>
> **Q3:**
> > The performance of ER and ExpoComm are very smilar in the filled bar transferability cases (i.e. K=log2N). Do you have theories about why this is?
>
> **A3:** Thank you for raising this insightful point! We also observed this phenomenon and included a brief discussion in Section 4.2 (lines 476–479). The strong transferability of both ER and ExpoComm shall result from the global grounding of messages, a design choice we implemented for both our proposed ExpoComm and the ER baseline.
>
> -----
>
> References
>
> [1] Rundong Wang, Xu He, Runsheng Yu, Wei Qiu, Bo An, and Zinovi Rabinovich. Learning efficient multi-agent communication: An information bottleneck approach. In Proceedings of the 37th International Conference on Machine Learning, pp. 9908–9918, 2020.
>
> [2] Shengchao Hu, Li Shen, Ya Zhang, and Dacheng Tao. Learning multi-agent communication from graph modeling perspective. In Proceedings of the 12th International Conference on Learning Representations, 2024.
>
> [3] Evgenii Nikishin, Junhyuk Oh, Georg Ostrovski, Clare Lyle, Razvan Pascanu, Will Dabney, and Andr´e Barreto. Deep reinforcement learning with plasticity injection. In Advances in Neural Information Processing Systems, volume 36, 2024.

---

> > ### Comment · Reviewer_6QM3 · 2024-11-27
> > **Response**
> >
> > Thank you to the authors for their responses and updates on the paper.  I have no further questions. I remain inclined to recommend acceptance of this paper.

---

> > > ### Author Response · Authors · 2024-11-27
> > >
> > > Thank you for your kind response. We sincerely appreciate the time and effort you have invested in providing valuable feedback to help improve our work.

---

### Official Review · Reviewer_56ww · 2024-11-04

**Soundness:** 3
**Presentation:** 3
**Contribution:** 2
**Rating:** 5
**Confidence:** 3

**Summary:**

The work focuses on the improving communication in large scale multi-agent reinforcement learning. The authors propose using exponential topology as a communication pattern among agents. The authors show that this method can improve the multi-agent performance in a large scale environment using the MAgent and Infrastructure Management Planning environments.

**Strengths:**

- The author studies the problem of improving large scale MARL communication, which can be helpful for MARL research and deployment.
- The paper is well written and easy to follow. The toy example and figures presented is helpful for understanding.
- The proposed method seems to show improved performance for the AdversarialPursuit case.

**Weaknesses:**

- It is unclear how the proposed exponential graph can be helpful for improving agent communication under target scenario. Providing some theoretically analysis or motivating example would be helpful. The toy example is helpful, but random global communication is not considered there.
- Beside the exponential graph, the contribution of this work seems limited. Highlighting and clarifying the contributions of this work would be helpful for better understanding.
- Evaluation seems incomplete. Comparing the proposed method to the traditional broadcast communication methods can help make the claim more persuasive.

**Questions:**

- How does the proposed method performs compared to traditional mulitcast type of multi-agent communication method such as CommNet, as it seems more global communication is beneficial for the target scenarios?
- How does the proposed exponential graph compared with random selection of communication peers? (keeping the number of communication peers the same)?
- What are the limitation of this method? Based on the results in Figure 4, the benefit of the method is more obvious for AdversarialPursuit than for Battle. It seems it's only learning after for the Battle environments. Is it learning faster for Battle because there is less communication peers for each node?

---

> ### Author Response · Authors · 2024-11-21
> **(1/2)**
>
> Thank you for your constructive feedback. Regarding your questions and suggestions, we have updated the manuscripts accordingly (highlighted in blue) and would like to provide clarifications below. If you have any follow-up questions or comments, please let us know, and we will be happy to discuss further.
>
> **Q1:**
> > It is unclear how the proposed exponential graph can be helpful for improving agent communication under target scenario. Providing some theoretically analysis or motivating example would be helpful. The toy example is helpful, but random global communication is not considered there.
>
> **A1:** The proposed ExpoComm enhances agent communication by introducing an topology that propagates information among all agents effectively and at low cost. To support this:
> - We analyze the exponential graph properties in Section 3.1.3 (lines 251-292). Specifically, the effective information propagation is enabled by the small-diameter property ($\text{diameter}(\mathcal{G}^t) = \lceil \log_2{(N-1)} \rceil$) and the low cost is ensured by the small-size property ($\lvert \mathcal{E}^t \rvert = N$ for one-peer exponential graph). Following suggestion from reviewer 56ww, we have supplemented the theoretical analysis in Appendix A to provide further support for these properties.
> - A toy example in Figure 2 illustrates message dissemination with different graph topologies. We demonstrate a trade-off between graph diameter and size, reflecting the balance between communication performance and overhead in many-agent systems. Exponential topologies strike a balance in this trade-off, showing strong information diffusion even with a minimal communication budget of $N$.
> - Extensive experimental results on benchmarks MAgent and IMP in Figure 4 and Table 1 show that ExpoComm outperforms baseline algorithms with the same communication budgets, demonstrating its effectiveness.
>
> To better address the reviewer’s concerns and clear up any misunderstanding, could reviewer 56ww kindly clarify what "random global communication" refers to in the original review?
>
> **Q2:** Beside the exponential graph, the contribution of this work seems limited. Highlighting and clarifying the contributions of this work would be helpful for better understanding.
>
> **A2:** Thank you for this suggestion. This work addresses a research gap in scalable multi-agent communication, as most existing strategies are designed for and tested under small-scale systems. Many real-world applications [1,2,3] require communication strategies that scale to dozens or even hundreds of agents. To address this gap, we made the following contributions:
> - We propose an exponential topology-enabled communication protocol, ExpoComm, as a scalable solution for MARL communication. It supports effective message dissemination among agents at low cost, enabled by the small-size and small-diameter properties of exponential graphs.
> - To fully leverage these properties for efficient information dissemination, we employ memory-based blocks for message processing and auxiliary tasks to ground messages, ensuring they effectively reflect global information.
> - Through extensive experiments across twelve scenarios on large-scale benchmarks, including MAgent and Infrastructure Management Planning (IMP), we demonstrate the superior performance and transferability of ExpoComm over existing baseline methods, handling large numbers of agents up to a hundred.
>
> **Q3:** Evaluation seems incomplete. Comparing the proposed method to the traditional broadcast communication methods can help make the claim more persuasive.
>
> **A3:** For comparisons with broadcast communication methods, please refer to A4 below. Regarding multicast communication, our original manuscript compares ExpoComm with traditional multicast methods, including CommFormer, ER, DGN+TarMAC, using two different communication budgets ($K = \lceil \log_2{N} \rceil$ and $K=1$). The results in Figure 4 and Table 1 demonstrate the superior performance of ExpoComm compared to these baselines. Please see Section 4.2 for a more detailed discussion.

---

> ### Author Response · Authors · 2024-11-21
> **(2/2)**
>
> **Q4:** How does the proposed method performs compared to traditional mulitcast type of multi-agent communication method such as CommNet, as it seems more global communication is beneficial for the target scenarios?
>
> **A4:** Thank you for the question, and we would like to provide the following clarifications:
> - Unlike ExpoComm, CommNet requires a physical communication proxy to aggregate messages from all agents during execution. This introduces additional hardware requirements for MASs, which is beyond the focus of our work. Without such a proxy, CommNet incurs $N$ (20 to 100) times more communication overhead compared to the $K=1$ case in our experiments. As a result, a direct comparison with CommNet is inherently unfair to ExpoComm and the other baseline methods.
> - However, we recognize that this comparison can provide useful insights into the properties of different tasks. And we have included these comparisons in Figure 11 and Table 6 in Appendix C.2. Despite incurring significantly lower communication costs, ExpoComm outperforms CommNet in most scenarios. Interestingly, CommNet shows comparable performance to ExpoComm in AdversarialPursuit tasks, suggesting that a global perspective is crucial in this scenario. This protentially explains the larger performance gap between ExpoComm and other baselines in this scenario.
> - Unlike global communication strategies that rely on physical proxies (e.g., CommNet), ExpoComm achieves global communication in decentralized MASs through a carefully designed communication topology. This emphasizes the versatility and scalability of ExpoComm, as it can achieve effective communication without the need for centralized infrastructure.
>
> **Q5:** How does the proposed exponential graph compared with random selection of communication peers? (keeping the number of communication peers the same)?
>
> **A5:** This corresponds to the ER baseline in our manuscript, with results presented in Figure 4 and Table 1. Overall, ExpoComm outperforms the ER method in most scenarios. Please see Section 4.2 for a more detailed discussion.
>
> **Q6:** What are the limitation of this method? Based on the results in Figure 4, the benefit of the method is more obvious for AdversarialPursuit than for Battle. It seems it's only learning after for the Battle environments. Is it learning faster for Battle because there is less communication peers for each node?
>
> **A6:** Thank you for your questions. We share our thoughts regarding these questions here:
> - Limitations: We have updated the manuscript with a subsection discussing limitations and future work (see Appendix C.3). We acknowledge that ExpoComm may not perform well in scenarios requiring more targeted communication, network MDPs, or non-cooperative tasks. We also suggest possible paths to further improve communication performance in many-agent systems.
> - Performance in AdversarialPursuit: The larger performance gap may be due to the stronger need for global information in AdversarialPursuit. This is supported by the superior performance of CommNet (see A4) and visualization results in Figure 9. In these tasks, agents move slower than adversaries, requiring more coordinated behaviors and a global perspective to trap adversaries effectively. ExpoComm provides this global perspective, making it particularly well-suited for tasks that demand strong coordination.
> - We are not entirely sure we fully understand the remaining part of the question. Could reviewer 56ww clarify what "it's only learning after for the Battle environments" means and what "it" refers to in the next sentence?
>
> We hope these responses and additional experiments address your concerns and encourage you to consider a more favorable evaluation of our paper.
>
> ----
>
> References
>
> [1] Kai Cui, Anam Tahir, Gizem Ekinci, Ahmed Elshamanhory, Yannick Eich, Mengguang Li, and Heinz Koeppl. A survey on large-population systems and scalable multi-agent reinforcement learning. arXiv preprint arXiv:2209.03859, 2022.
>
> [2] Lukas M Schmidt, Johanna Brosig, Axel Plinge, Bjoern M Eskofier, and Christopher Mutschler. An introduction to multi-agent reinforcement learning and review of its application to autonomous mobility. In IEEE 25th International Conference on Intelligent Transportation Systems (ITSC), pp. 1342–1349. IEEE, 2022.
>
> [3] Chengdong Ma, Aming Li, Yali Du, Hao Dong, and Yaodong Yang. Efficient and scalable reinforcement learning for large-scale network control. Nature Machine Intelligence, pp. 1–15, 2024.

---

> > ### Author Response · Authors · 2024-11-25
> >
> > As the rebuttal period is coming to an end, we would like to thank you again for your valuable feedback. In our rebuttal, we have:
> >
> > - Extended the theoretical analysis in Appendix A.
> > - Added a comparison with the proxy-based method CommNet in Figure 11 and Table 6 (Appendix C.2).
> > - Included a discussion on the limitations of our approach in Appendix C.3.
> > - Addressed and clarified the other questions raised in your review.
> >
> > We hope that our responses, along with the improvements in the revised manuscript, have sufficiently addressed your concerns. If this is the case, we would greatly appreciate it if you could consider updating your review score. If there are any remaining questions or concerns, please do not hesitate to let us know. Thank you again for your time and insights.

---

### Official Review · Reviewer_9jpX · 2024-11-05

**Soundness:** 3
**Presentation:** 3
**Contribution:** 4
**Rating:** 8
**Confidence:** 4

**Summary:**

The paper introduces ExpoComm, a scalable communication protocol for multi-agent reinforcement learning that uses exponential graph topologies to efficiently manage information flow among numerous agents in large-scale environments. Exponential graphs offer a small-diameter structure enabling ExpoComm to have rapid and cost-effective communication across agents. The method overcomes the need to finding task specific pairwise communication. The authors utilize a memory based message network for processing messages over time and to allow agents to accumulate and utilize past information. In addition, auxiliary tasks are used to align messages with global information, either through direct access to the global state (when available) or through contrastive learning techniques. The method is evaluated on MAgent gridworld benchmarks and compared with baselines having varying communication protocols.  The one-peer verison of the exponential graph performed the best despite only requiring a linear scaling communication cost.

**Strengths:**

The use of exponential graphs as a communication topology in MARL is an innovative approach.  Combining memory-based message processing and auxiliary tasks to enhance message relevance is also a strong contribution. The authors have performed extensive experiments with multiple baselines. The zero-shot transferability demonstrates a level of generalization. The method’s scalability and efficiency in managing communication in large agent populations without sacrificing performance have implications for large-scale MARL applications. Overall, the manuscript is well-organized and clearly presents both the motivation and implementation details of the proposed approach.

**Weaknesses:**

The paper would benefit from a more explicit discussion of the limitations of ExpoComm. Specifically, scenarios where the proposed exponential topology may not be ideal---such as tasks requiring task-specific pairwise communication links or settings where agents are non-cooperative or adversarial---are not fully addressed.

**Questions:**

The proposed exponential topology works well for cooperative tasks. However, could the authors clarify how this approach might be adapted for tasks where agents require specific pairwise connectivity? For instance, if a task necessitates more targeted information sharing between certain agents due to task-specific roles, would ExpoComm accommodate such requirements?

How would the proposed protocol perform in scenarios where some agents are adversarial or non-cooperative?

Could the authors provide more details about the experimental setups? Including environment visualizations or schematic diagrams would be extremely helpful for understanding the experimental conditions. Such visual aids could illustrate the setup, agent interactions, and communication patterns in more detail.

In Figures 4 and 6, the x-axis is labeled as "test return," although it appears to show plots related to training return. Could the authors clarify this discrepancy?

---

> ### Author Response · Authors · 2024-11-21
>
> Thank you for your thoughtful review. We have updated the manuscript accordingly (highlighted in blue) and provide clarifications below. If you have any follow-up questions or comments, please let us know, and we will be happy to discuss further.
>
> **Q1:** A more explicit discussion of the limitations: tasks reqiuring more targeted communication or non-cooperative tasks.
>
> **A1:** Thank you for the suggestion. We have included a subsection discussing limitations and future work (see Appendix C.3).
> - We acknowledge that ExpoComm may not perform well in scenarios requiring more targeted communication, network MDPs, or non-cooperative tasks. We have identified possible paths to further improve communication performance in many-agent systems.
> - Regarding the specific pairwise connectivity mentioned by reviewer 9jpX, this is indeed a very interesting question. Generally, there is a trade-off between adopting a global or local (pairwise) perspective when designing communication strategies in MASs of different scales. A local perspective, which focuses on task-oriented pairwise connectivity, can enhance task-specific performance in small-scale MASs, as shown in previous work [1]. However, it becomes extremely challenging to learn such relationships in large-scale MASs because the number of communication pairs scales quadratically with the number of agents. This observation motivates our ExpoComm, which adopts a global perspective in designing communication topology. While ExpoComm performs well in large-scale many-agent systems, it may not excel in scenarios requiring highly targeted communication. A promising direction for future work would be to design a mechanism that enables a seamless transition between global and local perspectives. Such a mechanism could potentially improve the adaptability of multi-agent communication schemes, allowing them to perform effectively across a wider range of scenarios. We leave this exploration for future work.
>
> **Q2:** More details and visualization about the experimental setups.
>
> **A2:** Thank you for the helpful suggestion! We have made the following updates to address this point:
> - Expanded Appendix B.2 to include more descriptions of the environmental settings along with snapshots (Figures 7 and 8) to improve clarity and readability.
> - Supplemented visualization results for ExpoComm and IDQN (without communication) in both AdversarialPursuit and Battle scenarios in Appendix C.1. These visualizations demonstrate that ExpoComm enhances cooperation by enabling agents to adopt a global perspective. For example, agents exhibit behaviors such as surrounding opponents and allocating more agents to the front lines, even when communication budgets are low ($K=1$). Please see Figures 9, 10, and Appendix C.1 for visualization results and more detailed discussion.
>
> **Q3:**
> > In Figures 4 and 6, the x-axis is labeled as "test return," although it appears to show plots related to training return. Could the authors clarify this discrepancy?
>
> **A3:** Thank you for pointing this out. To avoid any ambiguity, we have updated all y-axis labels to "evaluation return" or "evaluation win rate" (see Figures 4, 6, and 11). To clarify, the y-axis represents the evaluation return during the training process, which is induced by the learned policy without exploration (taking the argmax of Q functions), while the training return is induced by the learned policy with exploration (using epsilon-greedy strategies).
>
> -----
>
> References
>
> [1] Shengchao Hu, Li Shen, Ya Zhang, and Dacheng Tao. Learning multi-agent communication from graph modeling perspective. In Proceedings of the 12th International Conference on Learning Representations, 2024.

---

> > ### Comment · Reviewer_9jpX · 2024-11-26
> >
> > Thank you for the authors' response and the updates to the paper. Including the limitations enhances the reader's understanding, and the added images and details about the experimental setup improve clarity. I recommend revising the paper to address any remaining typographical and grammatical errors.

---

> > > ### Author Response · Authors · 2024-11-27
> > >
> > > We are pleased to hear that your initial concerns have been addressed, and we will continue proofreading the paper to resolve any remaining typos. Once again, we sincerely thank you for your time and effort in providing valuable feedback to help us improve our work.

---

### Official Review · Reviewer_MSLo · 2024-11-08

**Soundness:** 2
**Presentation:** 2
**Contribution:** 2
**Rating:** 6
**Confidence:** 2

**Summary:**

This paper proposes to utilize the exponential topology to enable rapid information dissemination among agents, which leads to scalable communication protocol ExpoComm. A memory-based message processors are employed and auxiliary loss is introduced to ground message. Experiments are conducted to validate their algorithm.

**Strengths:**

- Extensive experiments
- Superior performance against other baselines

**Weaknesses:**

- The presentation of section 3.3 is not ideal, e.g. line 323, can you elaborate more on the prediction function f? line 354 to line 355, what is t'?

**Questions:**

- In figure 4(f), why does the test win rate of ExpoComm start to drop from 4*1e6 step?
- This algorithm should excel in large-scale multi-agent environment, so can we suppose the performance gap between the algorithm proposed and other baselines should increase when the number of agents increase? If so, why can't we see such trend in figure 4?

---

> ### Author Response · Authors · 2024-11-21
>
> Thank you for your positive review. Regarding your questions and suggestions, we have updated the manuscripts accordingly (highlighted in blue) and would like to provide clarifications below. If you have any follow-up questions or comments, please let us know, and we will be happy to discuss further.
>
> **Q1:**
> >The presentation of section 3.3 is not ideal, e.g. line 323, can you elaborate more on the prediction function f? line 354 to line 355, what is t'?
>
> **A1:**
> Thank you for your suggestion. We have improved the readability of Section 3.3 by adding subtitles and rewording the relevant content. Please see the updated section highlighted in blue. The function $f(\cdot; \phi)$ refers to the learnable prediction function used for grounding messages. It is implemented using a two-layer MLP in our experiments (Appendix B.1, line 809). In Equation 5, $t$ and $t'$ refer to the timesteps of negative data pairs.
>
> **Q2:**
> > In figure 4(f), why does the test win rate of ExpoComm start to drop from 4*1e6 step?
>
> **A2:**
> We acknowledge that there is some performance fluctuation for ExpoComm throughout training in the Battle scenario. Similar fluctuations are also observed for baselines in the same scenario, such as IDQN (green solid line) at $4.5 \times 10^6$ timesteps in Battle w/ 64 agents, DGN+TarMAC (green solid line) at $2.6 \times 10^6$ timesteps in Battle w/ 20 agents and CommFormer (purple solid line) at $4.3 \times 10^6$ timesteps in Battle w/ 20 agents. We suspect this fluctuation may be due to the higher sensitivity of the win rate metric compared to return, as there were no outstanding abnormal patterns in return or loss curves for ExpoComm or the baselines.
>
> **Q3:**
> > This algorithm should excel in large-scale multi-agent environment, so can we suppose the performance gap between the algorithm proposed and other baselines should increase when the number of agents increase? If so, why can't we see such trend in figure 4?
>
> **A3:** To some extent, this is true. Such implications can be observed from the AdversarialPursuit scenario in Figure 4, Uncorrelated and Correlated scenario in Table 1. However, we refrained from explicitly making this claim in our manuscript due to the challenges in rigorously defining the performance gap across different settings. Specifically:
> - Comparing returns (or their differences) directly across different settings is problematic, as the returns are defined under different conditions.
> - It is unclear which baseline should be selected as the reference algorithm when calculating such a performance gap.

---

> > ### Comment · Reviewer_MSLo · 2024-11-26
> >
> > Thank you to the authors for their responses and updates on the paper. Everything is clear now, and I have no further questions. I remain inclined to recommend acceptance of this paper.

---

> > > ### Author Response · Authors · 2024-11-27
> > >
> > > We are delighted to hear that you now find everything clear, and we sincerely appreciate your support. Once again, thank you for your time and effort in helping us improve our work.

---

### Author Response · Authors · 2024-12-02
**Summary**

We sincerely thank all reviewers for their insightful comments and valuable feedback.

In this work, we address the challenge of scalable communication in multi-agent reinforcement learning (MARL) and introduce ExpoComm, an exponential topology-enabled communication protocol. Our framework leverages communication topologies with small diameters for fast information dissemination and small graph sizes for reduced communication overhead. This design enables effective and scalable communication strategies that achieve superior performance and strong transferability, while maintaining (near-)linear communication costs relative to the number of agents.

We are encouraged by the reviewers’ recognition of various aspects of our work. Specifically, we are pleased that our research question was considered **helpful for MARL research** (56ww, 6QM3), our method was recognized as **innovative** (9jpX, 6QM3) and **well-motivated** (9jpX), our experiments were regarded as **extensive** (MSLo, 9jpX, 6QM3), and our presentation was found **well-organized** (9jpX, 6QM3) and **easy to follow** (56ww).

In response to the reviewers' comments and suggestions, we have provided detailed point-by-point responses and made the following key updates to the manuscript:
- **Explicit definition of communication costs** in Section 3.1.1 and Section 4.1 to enhance clarity and readability
- **Theoretical analysis** in Appendix A to support the small-diameter property of exponential topologies
- **Detailed descriptions of environmental settings** in Appendix B.2 to improve the clarity
- **Visualization results** in Appendix C.1 to illustrate the cooperation patterns induced by the proposed communication strategies
- **Experimental comparison with CommNet** in Appendix C.2 to demonstrate the superior performance and lower cost of ExpoComm compared to proxy-based communication methods
- **Discussion on limitations** in Appendix C.3 to highlight potential future research directions

**During the rebuttal period, we believe we adequately addressed all questions and concerns raised by reviewers.** We are grateful that reviewers MSLo, 9jpX, and 6QM3 acknowledged the improvements made to the manuscript. We sincerely thank the reviewers, ACs, SACs, and PCs for their time and efforts in evaluating our work.

---

### Meta-Review · Area_Chair_7zdC · 2024-12-22

**Metareview:**

This paper studies the very timely and important problem of efficient communication in multi-agent RL. This is an under-studied topic in the RL community that deserves more attention. The proposed approach based on exponential topology is refreshing and brings new connections that likely will likely spawn interesting follow-up works. Particularly, the proposed framework leverages communication topologies with small diameters for fast information dissemination and small graph sizes for reduced communication overhead. This design allows the system to maintain roughly linear communication costs relative to the number of agents, and serves as a decent baseline for follow-up works.

**Additional Comments On Reviewer Discussion:**

NA

---

### Decision · Program_Chairs · 2025-01-22

Accept (Poster)